# Fe-catalyzed Decarbonylative Alkylative Spirocyclization of *N*-Arylcinnamamides: Access to Alkylated 1-Azaspirocyclohexadienones

**DOI:** 10.3390/molecules25030432

**Published:** 2020-01-21

**Authors:** Xiang Peng, Ren-Xiang Liu, Xiang-Yan Xiao, Luo Yang

**Affiliations:** Key Laboratory for Environmentally Friendly Chemistry and Application of the Ministry of Education, Key Laboratory for Green Organic Synthesis and Application of Hunan Province, College of Chemistry, Xiangtan University, Hunan 411105, China; pxsb123456@163.com (X.P.); lrx201821531602@163.com (R.-X.L.); xxy0707@163.com (X.-Y.X.)

**Keywords:** decarbonylation, alkylation, spirocyclization, aldehyde, cinnamamide

## Abstract

For the convenient introduction of simple linear/branched alkyl groups into biologically important azaspirocyclohexadienones, a practical Fe-catalyzed decarbonylative cascade spiro-cyclization of *N*-aryl cinnamamides with aliphatic aldehydes to provide alkylated 1-azaspiro-cyclohexadienones was developed. Aliphatic aldehydes were oxidative decarbonylated into primary, secondary and tertiary alkyl radicals conveniently and allows for the subsequent cascade construction of dual C(sp^3^)-C(sp^3^) and C=O bonds via radical addition, spirocyclization and oxidation sequence.

## 1. Introduction

The azaspirocyclohexadienone ring represents an important structural motif in natural compounds and pharmaceuticals, such as Annosqualine, Erythratinone and compound **I** (acting as muscarinic antagonist, Figure 1) [1,2,3,4]. Thus, the development of efficient methods for the synthesis of these azaspirocyclohexadienones has attracted substantial attention. While traditional synthetic routes rely predominantly on the transition-metal-catalyzed intramolecular *ipso*-carbocyclization [5,6,7,8,9] and electrophilic *ipso*-cyclization [10,11,12], the radical cascade *ipso*-cyclization was expanding in recent years, to incorporate various functional groups into the azaspirocyclohexadienone framework [13,14,15]. Among them, the radical difunctionalization and *ipso*-cyclization of *N*-arylcinnamamides or *N*-arylpropiolamides has proven to be an straightforward pathway, where various carbon or heteroatom centered radicals added onto the α,β-unsaturated carbon-carbon multiple bond of substrates, followed by the intramolecular radical *ipso*-cyclization and dearomatization. In this context, simple ethers [16], alkanes [17,18], ketones [19], acetonitrile [20], acyl chloride [21], aldehydes [22], aryldiazonium salts [23], CF_3_SO_2_Na [24,25,26], arylsulfinic acids [27], sulfonylhydrazides [28], AgSCF_3_ [29], thiophenols [30], disulfides [31], *N*-sulfanylsuccinimide [32], diselenide [33], *tert*-butyl nitrite [34], phosphonates [24,35,36] and silanes [37,38] have been demonstrated as the radical precursors for this spirocyclization (Scheme 1a), as independently reported by Li [16,17], Liang [25,29], and Zhu [18,20], etc. The addition of carbon radicals to *N*-arylcinnamamides and subsequent *ipso*-cyclization offers a direct synthesis of azaspiro-compounds with the concurrent construction of two C–C single bonds. However, the alkyl radical source for this spirocyclization of alkenes is still confined. When ethers, alkanes, ketones and acetonitrile acted as carbon radical precursors via the homolytic cleavage of sp^3^ C-H bond, the functional groups (ether, ketone or cyano groups) would inevitably be imported into the products, and the generation of regio-isomers was also troublesome due to the possible existence several different sp^3^ C-H bonds in these precursors, so readily available alkyl precursors that could introduce ordinary and simple linear/branched alkyl groups into azaspirocyclohexadienones are highly desirable, especially those could realize the convenient radical generation and be compatible with the primary, secondary and tertiary alkyls.

On the other hand, aldehydes are cheap and readily available chemicals and have been directly used for decarbonylative couplings catalyzed by ruthenium or rhodium, as shown by the extensive studies of Li since 2009 [39,40,41,42,43,44]. In contrast, we are interested in the radical-type decarbonylative reactions of aldehydes in the absence of noble metals, with peroxides as radical initiator and oxidant [45,46,47,48,49,50,51,52,53]. The oxidative decarbonylative couplings of aldehydes with (hetero) arenes [45,46], styrenes [47,48,49,50], alkyne and electron-deficient alkenes [51,52,53] were successively developed by our group. These decarbonylative reactions were further updated by other groups, with dioxygen as the radical initiator and oxidant [54,55,56]. Similar radical type decarbonylative alkylations of C=C and C≡C bonds with aldehydes were also separately developed by Li [57,58], Li [59,60,61,62] and Yu [63,64].

The above studies have fully demonstrated that radical-type decarbonylation of aliphatic aldehydes was an economic and convenient way to obtain primary, secondary and tertiary alkyl radicals, thus we postulated that the merging the oxidative decarbonylation of aliphatic aldehydes into the radical difunctionalization of *N*-arylcinnamamides, would produce a benzyl radical and then facilitate the subsequent radical *ipso*-cyclization (Scheme 1b). Herein, we report a novel Fe-promoted oxidative decarbonylative alkylative spirocyclization of *N*-arylcinnamamides with aliphatic aldehydes to provide alkylated 1-azaspirocyclohexadienones.

## 2. Results

Based on the above speculation, *N*-(4-hydroxyphenyl)-*N*-methyl cinnamamide (**1a**) and isobutyraldehyde (**2a**) were chosen as the model substrates for this oxidative decarbonylative spirocyclization; with di-*tert*-butyl peroxide (DTBP) as the radical initiator and terminal oxidant, the desired alkylated 1-azaspirocyclohexadienone **3a** was isolated and characterized. Detailed optimization was performed focusing on different iron salt and its loading, the dosage of aldehyde and DTBP, the reaction temperature and reaction solvent, which revealed the combination of 2.5 mol% Fe(acac)_2/5_ equiv aldehyde/3 equiv DTBP proved to be the most effective one, to afford the spirocyclization product **3a** in 67% isolated yield (Table 1, entry 1). For the reaction solvent, low polarity solvents including chlorobenzene (PhCl), trifluoromethylbenzene (PhCF_3_) and much more polar acetonitrile, ethyl acetate (EA) all turned out to be compatible, and among these solvents tested, ethyl acetate provided the best result (entries 12–14). We are glad that this radical cascade reaction favors ethyl acetate, which should be the greenest solvent among the ordinary organic solvents.

With the optimized reaction conditions in hand, we first examined the generality of this alkylative spirocyclization with different aliphatic aldehydes (**2a**–**2l**, Table 2). Various α-mono-substituted aliphatic aldehydes including isobutyraldehyde (**2a**), 2-ethylbutanal (**2b**), 2-methyl-butanal (**2c**), 2-methylpentanal (**2d**), 2-ethylhexanal (**2e**), cyclamen aldehyde (**2f**), cyclohexane-carbaldehyde (**2g**) and cyclopantanecarbaldehyde (**2h**) provided the corresponding secondary carbon radicals for the cascade spirocyclization after the oxidative decarbonylation. While the α-di-substituted pivaldehyde (**2i**) provided a tertiary carbon radical, the α-unsubstituted aliphatic aldehyde 3-methylbutanal (**2j**), propionaldehyde (**2k**) and 2-phenylacetaldehyde (**2l**) would provide primary carbon radicals, similarly. Gratifyingly, all of these aliphatic aldehydes underwent this decarbonylative alkylative spirocyclization witH-*N*-arylcinnamamide (**1a**) to produce the targeted alkylated 1-azaspirocyclohexadienone (**3a**–**3l**) smoothly. Moreover, the introduced alkyl group and the aryl group exhibit a trans-configuration, determined by the adjacent coupling constant of the proton on C3 and C4 (*J* = 12.0 Hz), which agreed well with the literature reports [18]. Considering the readily availability of these aliphatic aldehydes, avoidance the possible carbon radical rearrangement (to provide regioisomers as the alkane C-H bonds homolytic cleavage products) and simple operation for the radical generation (overriding pre-functionalization and photoreaction devices), this decarbonylative alkylative spirocyclization demonstrated again that the aliphatic aldehydes were convenient primary, secondary and tertiary alkyl precursors.

After investigating the scope of aliphatic aldehydes, we next tested the generality of this decarbonylative alkylative spirocyclization on different *N*-aryl cinnamamides **1b**–**1k** under the optimized conditions. The effect of substituents on the cinnamamide moiety is listed in Figure 2.

Various electron withdrawing or donating substituents were successfully incorporated into the cinnamamide unit of substrates **1b**–**1g**, such as methyl, halo and trifluoromethyl groups. Among them, the optimized reaction conditions were compatible with the cinnamamides with chloro groups substituted at the *para*, *meta* and *ortho* positions (compounds **1c**–**1e**), and similar yields were obtained, which revealed the substituents didn’t cause obvious steric hindrance for this cascade reaction. For the substituent on the *N*-linkage, the ethyl and benzyl group-substituted cinnamamides **1h** and **1i** provided slightly better yields than the model substrate **1a**. To our delight, the 2-naphthalenyl and 2-furanyl units (**1j** and **1k**) could also be introduced onto the α,β-unsaturated C=C bond of amide substrates, and the cascade reaction provided the 1-azaspirocyclohexadienone **4j** and **4k** in moderate yields.

Several control experiments were carried out to understand this decarbonylative alkylative spirocyclization. First, the cascade reaction of cinnamamide **1a** and aliphatic aldehyde **2a** was inhibited in the presence of di-*tert*-butylhydroxytoluene (BHT); instead, the decarbonylated alkyl radical was captured as 2,6-di-*tert*-butyl-4-isopropyl-4-methylcyclohexa-2,5-dien-1-one (**5**), which confirmed the radical-type decarbonylation mechanism (Scheme 2a). Second, the control experiment using the *N*-(4-methoxyphenyl)-*N*-methyl cinnamamide (**1l**) to replace the *N*-(4-hydroxyphenyl)-*N*-methyl cinnamamide (**1a**) was conducted under the optimized conditions, however no desired *5-exo*-trig spirocyclization product (**3a**) could be detected; in contrast, the C3-alkylated 3,4-dihydroquinolin-2(1*H*)-one **6** was formed in 74% yield, via 6-*endo*-trig cyclization pathway (Scheme 2b). The sharp difference on reactivity demonstrated the importance of the *para*-hydroxyl substituent for this spirocyclization, maybe due to its ability to stabilize the radical intermediate (**B**, Scheme 3) obtained from the 5-*exo*-trig spirocyclization and accelerating the subsequent cyclohexadienone formation.

Based on the mechanistic experiments and the previous studies [46,47,48,49,50,51,52,53], a possible reaction pathway is depicted in Scheme 3, with the reaction of *N*-(4-hydroxyphenyl)-*N*-methyl cinnamamide (**1a**) and isobutyraldehyde (**2a**) as an example.

First, promoted by the iron-catalyst, the homolytic cleavage of DTBP at elevated temperature forms *tert*-butoxy radical. Subsequent intermolecular hydrogen atom abstraction of the aldehyde (**2a**), spontaneous decarbonylation and insertion into the C=C bond of the cinnamamide (**1a**) affords a metastable benzyl radical **A**, which then adds onto the *ispo*-carbon to give the spiro-cyclohexadienyl radical **B**. The radical **B** is preferably oxidized by Fe^3+^ and deprotonated by a *tert*-butoxide anion to afford the 1-azaspirocyclohexadienone (**3a**).

## 3. Experimental

### 3.1. General Information

Unless otherwise noted, all commercially available compounds were used as provided without further purification. The substrates (various *N*-aryl cinnamamides) were synthesized from cinnamic acid and *para*-anisidine according to literature reports [18]. Dry solvents (toluene, ethyl acetate, dichloroethane, acetonitrile, chlorobenzene, fluorobenzene) were used as commercially available. Thin-layer chromatography (TLC) was performed using silica gel 60 F254 precoated plates (0.25 mm) or Sorbent Silica Gel 60 F254 plates (E. Merck). The developed chromatography was analyzed by UV lamp (254 nm). Unless other noted, high-resolution mass spectra (HRMS) were obtained from a JMS-700 instrument (ESI; JEOL). Melting points are uncorrected. Nuclear magnetic resonance (NMR) spectra were recorded on an Avance 400 spectrometer (Bruker) at ambient temperature. Chemical shifts for ^1^H-NMR spectra are reported in parts per million (ppm) from tetramethylsilane with the solvent resonance as the internal standard (chloroform: δ 7.26 ppm). Chemical shifts for ^13^C-NMR spectra are reported in parts per million (ppm) from tetramethylsilane with the solvent as the internal standard (CDCl_3_: δ 77.16 ppm). Data are reported as following: chemical shift, multiplicity (s = singlet, d = doublet, dd = doublet of doublets, t = triplet, q = quartet, m = multiplet, br = broad signal), coupling constant (Hz), and integration.

### 3.2. General Experimental Procedures

An oven-dried microwave reaction vessel was charged with FeCl_2_ (2.5 mol%) in EA (0.5 mL, pre-prepared solution), *N*-(4-hydroxyphenyl)-*N*-methylcinnamamide (**1a**, 0.1 mmol, 1.0 equiv), isobutyraldehyde (**2a**, 0.5 mmol, 5.0 equiv) and DTBP (0.3 mmol, 3.0 equiv). The vessel was sealed and heated at 122 °C (oil bath temperature) for 24 h. Afterwards the resulting mixture was cooled to room temperature, the solvent was removed in vacuo. The residue was purified by column chromatography on silica gel with a mixture of ethyl acetate/petroleum ether (1:3) as eluent to give the product **3a**.

### 3.3. Spectra Data of Products ***3a**–**3l***, ***4b**–**4k***, ***5***, ***6*** (see “Appendix A” for details)

*3-Isopropyl-1-methyl-4-phenyl-1-azaspiro [4.5]deca-6,9-diene-2,8-dione* (**3a**). The title compound was prepared according to the general procedure described above by the reaction between *N*-(4-hydroxyphenyl)-*N*-methylcinnamamide (**1a**) with isobutyraldehyde (**2a**), and purified by flash column chromatography as yellow oil (19.8 mg, 67%). ^1^H-NMR (400 MHz, CDCl_3_) δ 7.27–7.24 (m, 3H), 7.10 (dd, *J* = 7.6, 2.4 Hz, 2H), 6.78 (dd, *J* = 10.0, 3.2 Hz, 1H), 6.55 (dd, *J* = 10.2, 3.0 Hz, 1H), 6.39 (dd, *J* = 10.2, 2.0 Hz, 1H), 6.00 (dd, *J* = 10.2, 2.0 Hz, 1H), 3.43 (d, *J* = 12.0 Hz, 1H), 3.14 (dd, *J* = 11.8, 3.6 Hz, 1H), 2.73 (s, 3H), 2.38–2.34 (m, 1H), 1.01 (d, *J* = 6.8 Hz, 3H), 0.83 (d, *J* = 7.2 Hz, 3H). ^13^C-NMR (100 MHz, CDCl_3_) δ 184.46, 175.22, 149.40, 147.05, 135.04, 132.33, 131.51, 128.63, 128.21, 64.95, 50.84, 49.05, 28.07, 27.14, 20.14, 18.78. IR (cm^−1^): 3032, 2965, 2932, 2875, 1672, 1630, 1606, 1498, 1446, 1454, 1418, 1393, 1374, 1260, 1141, 1119, 1065, 991, 865, 794, 724, 700. HRMS: calcd. for C_19_H_21_NO_2_ Na^+^ [M + Na]^+^: 318.1465; Found: 318.1442.

*1-Methyl-3-(pentan-3-yl)-4-phenyl-1-azaspiro[4 .5]deca-6,9-diene-2,8-dione* (**3b**). The title compound was prepared according to the general procedure described above by the reaction between *N*-(4-hydroxyphenyl)-*N*-methylcinnamamide (**1a**) with 2-ethylbutanal (**2b**), and purified by flash column chromatography as yellow oil (23.0 mg, 71%). ^1^H-NMR (400 MHz, CDCl_3_) δ 7.26–7.24 (m, 3H), 7.09 (dd, *J* = 7.6, 2.8 Hz, 2H), 6.77 (dd, *J* = 10.0, 3.2 Hz, 1H), 6.60 (dd, *J* = 10.4, 3.2 Hz, 1H), 6.37 (dd, *J* = 10.0, 2.0 Hz, 1H), 6.00 (dd, *J* = 10.0, 2.0 Hz, 1H), 3.45 (d, *J* = 11.8 Hz, 1H), 3.31 (dd, *J* = 11.8, 2.6 Hz, 1H), 2.74 (s, 3H), 1.78–1.74 (m, 2H), 1.53–1.46 (m, 1H), 1.39–1.32 (m, 2H), 0.96 (t, *J* = 7.4 Hz, 3H), 0.80 (t, *J* = 7.3 Hz, 3H). ^13^C-NMR (100 MHz, CDCl_3_) δ 184.46, 175.15, 149.45, 146.94, 134.72, 132.36, 131.58, 128.68, 128.59, 128.28, 65.02, 51.48, 48.30, 34.68, 27.12, 26.47, 16.47, 12.44. IR (cm^−1^): 3032, 2961, 2931, 2875, 1692, 1672, 1630, 1454, 1419, 1392, 1376, 1260, 1173, 1141, 1119, 1066, 992, 865, 723,700, 662, 563. HRMS: calcd. for C_21_H_25_NO_2_ Na^+^ [M + Na]^+^: 346.1778; Found: 346.1753.

*3-(sec-Butyl)-1-methyl-4-phenyl-1-azaspiro[4.5]deca-6,9-diene-2,8-dione* (**3c**). The title compound was prepared according to the general procedure described above by the reaction between *N*-(4-hydroxyphenyl)-*N*-methylcinnamamide (**1a**) with 2-methylbutanal (**2c**), and purified by flash column chromatography as yellow oil (21.0 mg, 68%). ^1^H-NMR (400 MHz, CDCl_3_) δ 7.27–7.24 (m, 3H), 7.09 (d, *J* = 9.6, 2.4 Hz, 2H), 6.78 (dd, *J* = 10.0, 2.8 Hz, 1H), 6.54 (dd, *J* = 10.0, 3.2 Hz, 1H), 6.39 (dd, *J* = 10.4, 2.0 Hz, 1H), 6.00 (dd, *J* = 10.0, 2.0 Hz, 1H), 3.45 (d, *J* = 12.0 Hz, 1H), 3.20 (dd, *J* = 12.0, 2.8 Hz, 1H), 2.73 (s, 3H), 1.93–1.85 (m, 1H), 1.60–1.56 (m, 1H), 1.45–1.37 (m, 1H), 0.93 (t, *J* = 7.4 Hz, 3H), 0.85 (d, *J* = 7.0 Hz, 3H). ^13^C-NMR (100 MHz, CDCl_3_) δ 184.46, 175.15, 149.45, 146.94, 134.72, 132.36, 131.58, 128.68, 128.59, 128.28, 65.02, 51.48, 48.30, 34.68, 27.12, 26.47, 16.47, 12.44. IR (cm^−1^): 3059, 3032, 2962, 2932, 2875, 1672, 1630, 1498, 1454, 1419, 1392, 1375, 1260, 1172, 1143, 1065, 990, 865, 767, 700, 621. HRMS: calcd. for C_20_H_23_NO_2_ Na^+^ [M + Na]^+^: 332.1621; Found: 332.1605.

*1-Methyl-3-(pentan-2-yl)-4-phenyl-1-azaspiro[4.5]deca-6,9-diene-2,8-dione* (**3d**). The title compound was prepared according to the general procedure described above by the reaction between *N*-(4-hydroxyphenyl)-*N*-methylcinnamamide (**1a**) with 2-methylpentanal (**2d**), and purified by flash column chromatography as yellow oil (22.3 mg, 69%). ^1^H-NMR (400 MHz, CDCl_3_) δ 7.26–7.23 (m, 3H), 7.09 (dd, *J* = 7.6, 2.4 Hz, 2H), 6.76 (dd, *J* = 10.0, 2.8 Hz, 1H), 6.57 (dd, *J* = 10.0, 2.8 Hz, 1H), 6.38 (dd, *J* = 10.0, 2.0 Hz, 1H), 6.00 (dd, *J* = 10.0, 2.0 Hz, 1H), 3.44 (d, *J* = 10.8 Hz, 1H), 3.21 (dd, *J* = 11.6, 3.2 Hz, 1H), 2.74 (s, 3H), 2.30–2.23 (m, 1H), 1.37–1.23 (m, 2H), 1.16–1.01 (m, 2H), 0.96 (d, *J* = 6.9 Hz, 3H), 0.70 (t, *J* = 7.4 Hz, 3H). ^13^C-NMR (100 MHz, CDCl_3_) δ 184.48, 175.64, 149.44, 147.26, 146.96, 135.09, 132.34, 132.24, 131.58, 131.47, 128.66, 128.58, 128.33, 128.29, 128.23, 65.05, 50.43, 48.48, 48.25, 36.50, 35.77, 32.70, 27.24, 20.94, 20.67, 15.97, 14.31, 13.98. IR (cm^−1^): 3059, 3032, 2958, 2928, 2872, 1672, 1630, 1498, 1454, 1421, 1377, 1261, 1172, 1119, 1067, 990, 865, 794, 724, 700, 621. HRMS: calcd. for C_21_H_25_NO_2_ Na^+^ [M + Na]^+^: 346.1778; Found: 346.1752.

*3-(Heptan-3-yl)-1-methyl-4-phenyl-1-azaspiro[4.5]deca-6,9-diene-2,8-dione* (**3e**) [18]. The title compound was prepared according to the general procedure described above by the reaction between *N*-(4-hydroxyphenyl)-*N*-methylcinnamamide (**1a**) with 2-ethylhexanal (**2e**), and purified by flash column chromatography as yellow oil (21.4 mg, 61%). ^1^H-NMR (400 MHz, CDCl_3_) δ 7.25–7.22 (m, 3H), 7.09 (dd, *J* = 7.6, 2.4 Hz, 2H), 6.77 (dt, *J* = 10.0, 2.8 Hz, 1H), 6.60 (dt, *J* = 10.2, 3.2 Hz, 1H), 6.37 (dd, *J* = 10.0, 2.0 Hz, 1H), 6.00 (dt, *J* = 10.2, 2.2 Hz, 1H), 3.44 (dd, *J* = 11.8, 3.2 Hz, 1H), 3.30 (dd, *J* = 12.0, 3.0 Hz, 1H), 2.74 (s, 3H), 1.84–1.79 (m, 1H), 1.53–1.37 (m, 1H), 1.37–1.26 (m, 4H), 1.23–1.11 (m, 3H), 0.95 (t, *J* = 8.6 Hz, 3H), 0.81–0.72 (m, 3H). ^13^C-NMR (100 MHz, CDCl_3_) δ 184.48, 175.79, 149.53, 147.23, 134.93, 132.23, 131.54, 128.61, 128.44, 128.29, 65.08, 51.27, 46.04, 45.96, 40.21, 40.10, 31.12, 30.60, 30.31, 30.01, 27.22, 24.47, 24.24, 23.10, 22.80, 14.24, 14.02, 12.69, 12.24. IR (cm^−1^): 3060, 3032, 2958, 2929, 2872, 1691, 1672, 1630, 1499, 1455, 1393, 1376, 1259, 1173, 1119, 1066, 991, 865, 766,722, 700, 662.

(**3f**) *3-(1-(4-isopropylphenyl)propan-2-yl)-1-methyl-4-phenyl-1-azaspiro[4.5]deca-6,9-diene-2,8-dione*. The title compound was prepared according to the general procedure described above by the reaction between *N*-(4-hydroxyphenyl)-*N*-methylcinnamamide (**1a**) with 3-(4-isopropylphenyl)-2-methylpropanal (**2f**), and purified by flash column chromatography as yellow oil (30.6 mg, 74%). ^1^H-NMR (400 MHz, CDCl_3_) δ 7.24–7.23 (m, 1H), 7.17–7.13 (m, 2.5H), 7.20–7.07 (m, 2.5H), 7.00 (dd, *J* = 6.0, 2.0 Hz, 1H), 6.89 (dd, *J* = 6.0, 1.6 Hz, 1H), 6.80–6.73 (m, 1.5H), 6.65 (dd, *J* = 10.0, 3.2 Hz, 0.5H), 6.53 (dd, *J* = 11.4, 3.2 Hz, 0.5H), 6.43–6.32 (m, 1.5H), 5.96 (td, *J* = 10.1, 2.0 Hz, 1H), 3.44–3.38 (m, 1H), 3.22–3.14 (m, 1H), 3.08–3.02 (m, 1H), 2.91–2.84 (m, 1H), 2.72 (d, *J* = 8 Hz, 3H), 2.67 (t, *J* = 6.8 Hz, 0.5H), 2.49 (q, *J* = 6.8 Hz, 0.5H), 2.28 (d, 5.6 Hz, 0.5H), 2.06–2.00(m, 1H), 1.28–1.21 (m, 6H), 0.99 (dd, *J* = 6.8, 4.4 Hz, 3H). ^13^C-NMR (100 MHz, CDCl_3_) δ 174.77, 149.42, 149.36, 147.09, 146.84, 146.76, 146.58, 137.97, 137.48, 134.68, 133.85, 132.44, 132.18, 131.53, 131.45, 129.49, 129.06, 128.61, 128.58, 128.54, 128.30, 128.20, 128.00, 126.41, 126.33, 64.95, 64.89, 51.61, 50.78, 47.81, 45.16, 40.13, 40.02, 34.93, 34.56, 33.86, 33.79, 27.19, 26.98, 24.33, 24.19, 16.74, 15.80. IR (cm^−1^): 3049, 3030, 2960, 2928, 2873, 1689, 1631, 1499, 1454, 1392, 1378, 1265, 1172, 1114, 1058, 990, 864, 735, 700, 570. HRMS: calcd. for C_28_H_31_NO_2_ Na^+^ [M + Na]^+^: 436.2247; Found: 436.2217.

*3-Cyclohexyl-1-methyl-4-phenyl-1-azaspiro[4.5]deca-6,9-diene-2,8-dione* (**3g**) [18]. The title compound was prepared according to the general procedure described above by the reaction between *N*-(4-hydroxyphenyl)-*N*-methylcinnamamide (**1a**) with cyclohexanecarbaldehyde (**2g**), and purified by flash column chromatography as yellow oil (24.1 mg, 72%). ^1^H-NMR (400 MHz, CDCl_3_) δ 7.27–7.24(m, 3H), 7.09 (dd, *J* = 7.6, 2.4 Hz, 2H), 6.76 (dd, *J* = 10.0, 3.2 Hz, 1H), 6.54 (dd, *J* = 10.2, 3.0 Hz, 1H), 6.39 (dd, *J* = 10.4, 2.0 Hz, 1H), 5.98 (dd, *J* = 10.2, 2.0 Hz, 1H), 3.48 (d, *J* = 12.0 Hz, 1H), 3.11 (dd, *J* = 11.8, 3.6 Hz, 1H), 2.73 (s, 3H), 2.01–1.94 (m, 1H), 1.73 (d, *J* = 13.2, 1H), 1.63–1.56 (m, 3H), 1.38–1.13 (m, 4H), 1.08–0.99 (m, 1H), 0.87–0.81 (m, 1H). ^13^C-NMR (100 MHz, CDCl_3_) δ 184.50, 175.30, 149.45, 147.09, 135.06, 132.33, 131.45, 128.63, 128.22, 128.19, 65.00, 51.02, 48.81, 38.27, 30.98, 29.14, 27.19, 26.71, 26.56, 26.23. IR (cm^−1^): 3057, 3032, 2925, 2852, 1690, 1672, 1499, 1450, 1419, 1393, 1377, 1260, 1172, 1134, 1096, 1069, 991, 864, 796, 732, 657, 569.

*3-Cyclopentyl-1-methyl-4-phenyl-1-azaspiro[4.5]deca-6,9-diene-2,8-dione* (**3h**) [18]. The title compound was prepared according to the general procedure described above by the reaction between *N*-(4-hydroxyphenyl)-*N*-methylcinnamamide (**1a**) with cyclopentanecarbaldehyde (**2h**), and purified by flash column chromatography as yellow oil (19.6 mg, 61%). ^1^H-NMR (400 MHz, CDCl_3_) δ 7.27–7.24 (m, 3H), 7.09 (dd, *J* = 7.6, 2.4 Hz, 2H), 6.77 (dd, *J* = 10.2, 3.0 Hz, 1H), 6.54 (dd, *J* = 10.2, 3.1 Hz, 1H), 6.39 (dd, *J* = 10.2, 2.0 Hz, 1H), 5.99 (dd, *J* = 10.2, 2.0 Hz, 1H), 3.38 (d, *J* = 11.6 Hz, 1H), 3.21 (dd, *J* = 11.8, 6.2 Hz, 1H), 2.73 (s, 3H), 2.24–2.18 (m, 1H), 2.05–1.96 (m, 1H), 1.87–1.76 (m, 3H), 1.50–1.38 (m, 3H), 1.25–1.21 (m, 1H). ^13^C-NMR (100 MHz, CDCl_3_) δ 184.50, 175.30, 149.45, 147.09, 135.06, 132.33, 131.45, 128.63, 128.22, 128.19, 65.06, 53.45, 46.83, 41.20, 29.85, 29.56, 27.16, 25.18, 24.99. IR (cm^−1^): 3059, 3031, 2923, 2869, 1730, 1692, 1671, 1630, 1453, 1442, 1393, 1375, 1260, 1172, 1075, 991, 865, 794, 732,700, 645, 569.

*3-(tert-Butyl)-1-methyl-4-phenyl-1-azaspiro[4.5]deca-6,9-diene-2,8-dione* (**3i**). The title compound was prepared according to the general procedure described above by the reaction between *N*-(4-hydroxyphenyl)-*N*-methylcinnamamide (**1a**) with pivalaldehyde (**2i**), and purified by flash column chromatography as yellow oil (16.7 mg, 52%). ^1^H-NMR (400 MHz, CDCl_3_) δ 7.27 (s, 3H), 7.21 (s, 2H), 6.77 (dd, *J* = 10.0, 3.2 Hz, 1H), 6.46 (dd, *J* = 10.2, 3.2 Hz, 1H), 6.38 (dd, *J* = 10.0, 2.0 Hz, 1H), 5.92 (dd, *J* = 10.2, 2.0 Hz, 1H), 3.40 (d, *J* = 11.2 Hz, 1H), 3.01 (d, *J* = 11.6 Hz, 1H), 2.70 (s, 3H), 1.00 (s, 9H). ^13^C-NMR (100 MHz, CDCl_3_) δ 184.48, 175.09, 149.54, 147.98, 136.60, 132.11, 131.04, 128.02, 64.35, 52.50, 51.33, 33.87, 28.08, 27.22. IR (cm^−1^): 3031, 2958, 2870, 1688, 1672, 1630, 1605, 1468, 1420, 1392, 1370, 1260, 1244, 1171, 1119, 1093, 864, 794, 720, 735,700, 610, 563. HRMS: calcd. for C_20_H_23_NO_2_ Na^+^ [M + Na]^+^: 332.1621; Found: 332.1597.

*3-Isobutyl-1-methyl-4-phenyl-1-azaspiro[4.5]deca-6,9-diene-2,8-dione* (**3j**). The title compound was prepared according to the general procedure described above by the reaction between *N*-(4-hydroxyphenyl)-*N*-methylcinnamamide (**1a**) with 3-methylbutanal (**2j**), and purified by flash column chromatography as yellow oil (19.8 mg, 64%). ^1^H-NMR (400 MHz, CDCl_3_) δ 7.27–7.24 (m, 3H), 7.09 (dd, *J* = 7.6, 2.4 Hz, 2H), 6.76 (dd, *J* = 10.0, 3.2 Hz, 1H), 6.59 (dd, *J* = 10.4, 3.0 Hz, 1H), 6.39 (dd, *J* = 10.0, 2.0 Hz, 1H), 6.01 (dd, *J* = 10.2, 2.0 Hz, 1H), 3.25 (d, *J* = 11.6 Hz, 1H), 3.15 – 3.09 (m, 1H), 2.74 (s, 3H), 1.89–1.83 (m, 1H), 1.74–1.67 (m, 1H), 1.38–1.31 (m, 1H), 0.86 (d, *J* = 7.2 Hz, 3H), 0.80 (d, *J* = 6.4 Hz, 3H). ^13^C-NMR (100 MHz, CDCl_3_) δ 184.45, 176.38, 149.36, 146.81, 134.34, 132.40, 131.59, 128.66, 128.36, 128.25, 65.23, 56.35, 41.66, 41.10, 27.23, 25.20, 22.93, 22.37. IR (cm^−1^): 3056, 3033, 2956, 2927, 2869, 1692, 1672, 1630, 1467, 1454, 1393, 1376, 1262, 1172, 1137, 1080, 1060, 866, 790, 721, 700. HRMS: calcd. for C_20_H_23_NO_2_ Na^+^ [M + Na]^+^: 332.1621; Found: 332.1597.

3-*Ethyl-1-methyl-4-phenyl-1-azaspiro[4.5]deca-6,9-diene-2,8-dione* (**3k**). The title compound was prepared according to the general procedure described above by the reaction between *N*-(4-hydroxyphenyl)-*N*-methylcinnamamide (**1a**) with propionaldehyde (**2k**), and purified by flash column chromatography as yellow oil (16.1 mg, 56%). ^1^H-NMR (400 MHz, CDCl_3_) δ 7.27–7.24 (m, 3H), 7.10 (dd, *J* = 8.0, 2.4 Hz 2H), 6.79 (dd, *J* = 10.0, 3.2 Hz, 1H), 6.57 (dd, *J* = 10.2, 3.2 Hz, 1H), 6.41 (dd, *J* = 10.0, 2.0 Hz, 1H), 6.02 (dd, *J* = 10.2, 2.0 Hz, 1H), 3.34 (d, *J* = 12.0 Hz, 1H), 3.13–3.06 (m, 1H), 2.75 (s, 3H), 1.96–1.86 (m, 1H), 1.77–1.68 (m, 1H), 0.89 (t, *J* = 7.4 Hz, 3H). ^13^C-NMR (100 MHz, CDCl_3_) δ 184.46, 175.72, 149.32, 146.57, 134.25, 132.53, 131.70, 128.70, 128.34, 128.17, 65.14, 54.39, 44.64, 27.17, 23.01, 11.18. IR (cm^−1^): 3056, 3032, 2965, 2931, 2877, 1692, 1672, 1629, 1454, 1420, 1394, 1376, 1262, 1174, 1125, 1060, 988, 865, 719, 699, 659. HRMS: calcd. for C_18_H_19_NO_2_ Na^+^ [M + Na]^+^: 304.1308; Found: 304.1286.

*3-Benzyl-1-methyl-4-phenyl-1-azaspiro[4.5]deca-6,9-diene-2,8-dione* (**3l**). The title compound was prepared according to the general procedure described above by the reaction between *N*-(4-hydroxyphenyl)-*N*-methylcinnamamide (**1a**) with 2-phenylacetaldehyde (**2l**), and purified by flash column chromatography as yellow oil (14.4 mg, 42%). ^1^H-NMR (400 MHz, CDCl_3_) δ 7.24–7.16 (m, 6H), 7.07–7.04 (m, 2H), 6.99–6.97 (m, 2H), 6.56 (dd, *J* = 10.2, 3.0 Hz, 1H), 6.47 (dd, *J* = 10.0, 3.0 Hz, 1H), 6.30 (dd, *J* = 10.2, 2.0 Hz, 1H), 6.00 (dd, *J* = 10.2, 2.0 Hz, 1H), 3.42 (dt, *J* = 12.2, 5.4 Hz, 1H), 3.30–3.20 (m, 2H), 2.93 (dd, *J* = 13.6, 5.8 Hz, 1H), 2.71 (s, 3H). ^13^C-NMR (100 MHz, CDCl_3_) δ 184.41, 174.77, 149.24, 146.36, 137.36, 133.43, 132.39, 131.76, 129.98, 128.69, 128.47, 128.28, 128.27, 126.71, 65.01, 52.74, 45.16, 34.06, 27.23. IR (cm^−1^): 3060, 3029, 2922, 2853, 1693, 1672, 1630, 1496, 1453, 1393, 1377, 1261, 1172, 1130, 1092, 1075, 865, 788, 721, 699. HRMS: calcd. for C_23_H_21_NO_2_ Na^+^ [M + Na]^+^: 366.1465; Found: 366.1443.

*3-Isopropyl-1-methyl-4-phenyl-1-azaspiro[4.5]deca-6,9-diene-2,8-dione* (**4b**). The title compound was prepared according to the general procedure described above by the reaction between *N*-(4-hydroxyphenyl)-*N*-methyl-3-(p-tolyl)acrylamide (**1b**) with isobutyraldehyde (**2a**), and purified by flash column chromatography as yellow oil (22.2 mg, 72%). ^1^H-NMR (400 MHz, CDCl_3_) δ 7.05 (d, *J* = 7.6 Hz, 2H), 6.53 (d, *J* = 7.6 Hz, 2H), 6.77 (dd, *J* = 10.0, 3.2 Hz, 1H), 6.56 (dd, *J* = 10.2, 3.0 Hz, 1H), 6.38 (dd, *J* = 10.0, 2.0 Hz, 1H), 6.01 (dd, *J* = 10.2, 1.8 Hz, 1H), 3.40 (d, *J* = 11.6 Hz, 1H), 3.11 (dd, *J* = 12.0, 3.6 Hz, 1H), 2.73 (s, 3H), 2.36–2.31 (m, 1H), 2.28 (s, 3H), 1.01 (d, *J* = 6.8 Hz, 3H), 0.83 (d, *J* = 7.2 Hz, 3H). ^13^C-NMR (100 MHz, CDCl_3_) δ 184.60, 175.34, 149.56, 147.20, 137.93, 132.28, 131.95, 131.50, 129.33, 128.09, 65.05, 50.58, 49.13, 28.07, 27.14, 21.16, 20.13, 18.83. IR (cm^−1^): 3046, 3030, 2960, 2929, 2876, 1690, 1672, 1629, 1516, 1465, 1419, 1393, 1375, 1263, 1065, 992, 864, 736, 639, 575. HRMS: calcd. for C_20_H_23_NO_2_ Na^+^ [M + Na]^+^: 332.1621; Found: 332.1602.

*4-(4-Chlorophenyl)-3-isopropyl-1-methyl-1-azaspiro[4.5]deca-6,9-diene-2,8-dione* (**4c**). The title compound was prepared according to the general procedure described above by the reaction between 3-(4-chlorophenyl)-*N*-(4-hydroxyphenyl)-*N*-methylacrylamide (**1c**) with isobutyraldehyde (**2a**), and purified by flash column chromatography as yellow oil (22.7 mg, 69%). ^1^H-NMR (400 MHz, CDCl_3_) δ 7.24 (dd, *J* = 6.4, 2.0 Hz, 2H), 7.05 (dd *J* = 6.4, 2.0 Hz, 2H), 6.76 (dd, *J* = 10.2, 3.0 Hz, 1H), 6.54 (dd, *J* = 10.2, 3.2 Hz, 1H), 6.40 (dd, *J* = 10.0, 2.0 Hz, 1H), 6.05 (dd, *J* = 10.2, 2.0 Hz, 1H), 3.40 (d, *J* = 12.0 Hz, 1H), 3.08 (dd, *J* = 12.0, 3.6 Hz, 1H), 2.73 (s, 3H), 2.38–2.30 (m, 1H), 1.00 (d, *J* = 6.8 Hz, 3H), 0.82 (d, *J* = 6.8 Hz, 3H). ^13^C-NMR (100 MHz, CDCl_3_) δ 184.23, 174.90, 149.09, 146.66, 134.10, 133.70, 132.53, 131.86, 129.53, 128.91, 64.83, 50.23, 49.23, 28.06, 27.19, 20.29, 18.68. IR (cm^−1^): 3050, 2961, 2931, 2874, 1692, 1672, 1630, 1494, 1466, 1417, 1392, 1373, 1259, 1173, 1121, 1092, 1014, 866, 830, 736. HRMS: calcd. for C_19_H_20_ClNO_2_ Na^+^ [M + Na]^+^: 352.1075; Found: 352.1059.

*4-(3-Chlorophenyl)-3-isopropyl-1-methyl-1-azaspiro[4.5]deca-6,9-diene-2,8-dione* (**4d**). The title compound was prepared according to the general procedure described above by the reaction between 3-(3-chlorophenyl)-*N*-(4-hydroxyphenyl)-*N*-methylacrylamide (**1d**) with isobutyraldehyde (**2a**), and purified by flash column chromatography as yellow oil (21.1 mg, 64%). ^1^H-NMR (400 MHz, CDCl_3_) δ 7.25–7.20 (m, 2H), 7.10 (s 1H), 7.00 (dt, *J* = 7.0, 1.8 Hz, 1H), 6.76 (dd, *J* = 10.2, 3.2 Hz, 1H), 6.56 (dd, *J* = 10.2, 3.0 Hz, 1H), 6.42 (dd, *J* = 10.0, 2.0 Hz, 1H), 6.06 (dd, *J* = 10.2, 2.0 Hz, 1H), 3.40 (d, *J* = 12.0 Hz, 1H), 3.08 (dd, *J* = 12.0, 3.6 Hz, 1H), 2.73 (s, 3H), 2.38–2.30 (m, 1H), 1.01 (d, *J* = 6.8 Hz, 3H), 0.83 (d, *J* = 6.8 Hz, 3H). ^13^C-NMR (100 MHz, CDCl_3_) δ 184.22, 174.78, 148.99, 146.52, 137.35, 134.58, 132.62, 131.87, 129.96, 128.55, 128.35, 126.53, 64.76, 50.52, 49.20, 28.09, 27.17, 20.23, 18.75. IR (cm^−1^): 3056, 2961, 2931, 2874, 1691, 1672, 1630, 1468, 1422, 1392, 1375, 1261, 1173, 1119, 1082, 880, 852, 736, 690, 623. HRMS: calcd. for C_19_H_20_ClNO_2_ Na^+^ [M + Na]^+^: 352.1075; Found: 352.1053.

*4-(2-Chlorophenyl)-3-isopopyl-1-methyl-1-azaspiro[4.5]deca-6,9-diene-2,8-dione* (**4e**). The title compound was prepared according to the general procedure described above by the reaction between 3-(2-chlorophenyl)-*N*-(4-hydroxyphenyl)-*N*-methylacrylamide (**1e**) with isobutyraldehyde (**2a**), and purified by flash column chromatography as yellow oil (21.0 mg, 63%). ^1^H-NMR (400 MHz, CDCl_3_) δ 7.34–7.24 (m, 3H), 7.22–7.16 (m, 2H), 6.88 (dd, *J* = 10.2, 3.0 Hz, 1H), 6.68 (dd, *J* = 10.2, 3.2 Hz, 1H), 6.33 (dd, *J* = 10.2, 2.0 Hz, 1H), 6.12 (dd, *J* = 10.2, 2.0 Hz, 1H), 4.25 (d, *J* = 11.8 Hz, 1H), 3.04 (dd, *J* = 11.8, 3.6 Hz, 1H), 2.74 (s, 3H), 2.39–2.31 (m, 1H), 1.00 (d, *J* = 6.8 Hz, 3H), 0.76 (d, *J* = 7.0 Hz, 3H). ^13^C-NMR (100 MHz, CDCl_3_) δ 184.41, 174.88, 149.76, 146.58, 133.44, 131.83, 131.81, 130.45, 135.01, 129.37, 129.22, 126.68, 64.64, 51.18, 46.06, 28.23, 26.94, 20.50, 18.43. IR (cm^−1^): 3059, 2960, 2931, 2873, 1692, 1672, 1631, 1468, 1422, 1392, 1374, 1260, 1174, 1116, 1065, 1037, 864, 750, 736, 698. HRMS: calcd. for C_19_H_20_ClNO_2_ Na^+^ [M + Na]^+^: 352.1075; Found: 352.1054.

*4-(4-Bromophenyl)-3-isopropyl-1-methyl-1-azaspiro[4.5]deca-6,9-diene-2,8-dione* (**4f**). The title compound was prepared according to the general procedure described above by the reaction between 3-(4-bromophenyl)-*N*-(4-hydroxyphenyl)-*N*-methylacrylamide (**1f**) with isobutyraldehyde (**2a**), and purified by flash column chromatography as yellow oil (23.5 mg, 63%). ^1^H-NMR (400 MHz, CDCl_3_) δ 7.41–7.38 (m, 2H), 6.99 (dd, *J* = 6.4, 2.0 Hz, 2H), 6.75 (dd, *J* = 10.0, 3.2 Hz, 1H), 6.53 (dd, *J* = 10.2, 3.0 Hz, 1H), 6.40 (dd, *J* = 10.2, 2.0 Hz, 1H), 6.05 (dd, *J* = 10.2, 2.0 Hz, 1H), 3.39 (d, *J* = 12.0 Hz, 1H), 3.07 (dd, *J* = 12.0, 3.6 Hz, 1H), 2.73 (s, 3H), 2.38–2.30 (m, 1H), 0.99 (d, *J* = 7.0 Hz, 3H), 0.82 (d, *J* = 7.0 Hz, 3H). ^13^C-NMR (100 MHz, CDCl_3_) δ 184.22, 174.83, 149.09, 146.60, 134.26, 132.56, 131.90, 131.88, 129.86, 122.24, 64.73, 50.31, 49.19, 28.09, 27.19, 20.31, 18.70. IR (cm^−1^): 3048, 2960, 2929, 2873, 1693, 1671, 1630, 1491, 1466, 1392, 1374, 1260, 1173, 1120, 1010, 866, 827, 755, 629, 571. HRMS: calcd. for C_19_H_20_BrNO_2_ Na^+^[M + Na]^+^: 396.0570; Found: 396.0550.

*3-Isopropyl-1-methyl-4-(4-(trifluoromethyl)phenyl)-1-azaspiro[4.5]deca-6,9-diene-2,8-dione* (**4g**). The title compound was prepared according to the general procedure described above by the reaction between *N*-(4-hydroxyphenyl)-*N*-methyl-3-(4-(trifluoromethyl)phenyl)acrylamide (**1g**) with isobutyraldehyde (**2a**), and purified by flash column chromatography as yellow oil (23.6 mg, 65%). ^1^H-NMR (400 MHz, CDCl_3_) δ 7.54 (d, *J* = 7.8 Hz, 2H), 7.25 (t, *J* = 7.4 Hz, 2H), 6.79 (dd, *J* = 10.2, 3.2 Hz, 1H), 6.55 (dd, *J* = 10.2, 3.2 Hz, 1H), 6.42 (dd, *J* = 10.0, 2.0 Hz, 1H), 6.04 (dd, *J* = 10.2, 2.0 Hz, 1H), 3.49 (d, *J* = 11.8 Hz, 1H), 3.15 (dd, *J* = 12.0, 3.8 Hz, 1H), 2.74 (s, 3H), 2.39–2.31 (m, 1H), 1.01 (d, *J* = 7.0 Hz, 3H), 0.82 (d, *J* = 7.0 Hz, 3H). ^13^C-NMR (100 MHz, CDCl_3_) δ 184.05, 174.67, 148.90, 146.28, 139.43, 132.68, 131.97, 128.69, 125.73, 125.69, 64.68, 50.57, 49.24, 28.13, 27.17, 20.30, 18.72. ^19^F-NMR (376 MHz, CDCl3) δ 62.65 (s, 1F). IR (cm^−1^): 3050, 2962, 2933, 2876, 1692, 1672, 1631, 1468, 1422, 1392, 1375, 1327, 1166, 1124, 1069, 1017, 869, 853, 737, 602. HRMS: calcd. for C_20_H_20_F_3_NO_2_ Na^+^ [M + Na]^+^: 386.1338; Found: 386.1319.

(*1-Ethyl-3-isopropyl-4-phenyl-1-azaspiro[4.5]deca-6,9-diene-2,8-dione* (**4h**). The title compound was prepared according to the general procedure described above by the reaction between *N*-ethyl-*N*-(4-hydroxyphenyl)cinnamamide (**1h**) with isobutyraldehyde (**2a**), and purified by flash column chromatography as yellow oil (22.0 mg, 71%). ^1^H-NMR (400 MHz, CDCl_3_) δ 7.27–7.23 (m, 3H), 7.09 (dd, *J* = 8.0, 2.0 Hz, 2H), 6.82 (dd, *J* = 10.1, 3.1 Hz, 1H), 6.62 (dd, *J* = 10.2, 3.0 Hz, 1H), 6.37 (dd, *J* = 10.1, 2.0 Hz, 1H), 5.95 (dd, *J* = 10.2, 2.0 Hz, 1H), 3.42 (d, *J* = 12.0 Hz, 1H), 3.36–3.27 (m, 1H), 3.14 (dd, *J* = 12.0, 3.6 Hz, 1H), 3.10–3.01 (m, 1H), 2.38–2.30 (m, 1H), 1.13 (t, *J* = 7.2 Hz, 3H), 1.01 (d, *J* = 7.0 Hz, 3H), 0.82 (d, *J* = 7.2 Hz, 3H). ^13^C-NMR (100 MHz, CDCl_3_) δ 184.71, 175.02, 149.52, 148.09, 135.02, 131.86, 130.60, 128.61, 128.28, 128.23, 65.24, 51.25, 49.04, 36.86, 28.11, 20.17, 18.71, 15.29. IR (cm^−1^): 3057, 3033, 2962, 2934, 2874, 1678, 1629, 1498, 1454, 1402, 1376, 1310, 1262, 1140, 1125, 1064, 940, 866, 724, 700. HRMS: calcd. for C_20_H_23_NO_2_ Na^+^ [M + Na]^+^: 332.1621; Found: 332.1597.

*1-Benzyl-3-isopropyl-4-phenyl-1-azaspiro[4.5]deca-6,9-diene-2,8-dione* (**4i**). The title compound was prepared according to the general procedure described above by the reaction between *N*-benzyl-*N*-(4-hydroxyphenyl)cinnamamide (**1i**) with isobutyraldehyde (**2a**), and purified by flash column chromatography as yellow oil (27.8 mg, 75%). ^1^H-NMR (400 MHz, CDCl_3_) δ 7.27–7.23 (m, 3H), 7.22–7.18 (m, 5H), 7.04 (dd, *J* = 6.8, 3.0 Hz, 2H), 6.54 (dd, *J* = 10.4, 3.2 Hz, 1H), 6.47 (dd, *J* = 10.2, 3.0 Hz, 1H), 6.18 (dd, *J* = 10.0, 2.0 Hz, 1H), 5.81 (dd, *J* = 10.2, 2.0 Hz, 1H), 4.66 (d, *J* = 15.0 Hz, 1H), 4.07 (d, *J* = 14.8 Hz, 1H), 3.42 (d, *J* = 12.0 Hz, 1H), 3.22 (dd, *J* = 12.0, 3.8 Hz, 1H), 2.43–2.35 (m, 1H), 1.07 (d, *J* = 6.8 Hz, 3H), 0.85 (d, *J* = 6.8 Hz, 3H). ^13^C-NMR (100 MHz, CDCl_3_) δ 184.70, 175.31, 149.35, 147.30, 137.95, 134.66, 131.35, 130.71, 128.64, 128.60, 128.53, 128.30, 128.24, 127.78, 65.26, 51.27, 48.88, 45.36, 28.14, 20.11, 18.86. IR (cm^−1^): 3061, 3031, 2961, 2930, 2873, 1670, 1629,1496, 1454, 1399, 1264, 1179, 1065, 1011, 958, 930, 863, 792, 734, 700. HRMS: calcd. for C_25_H_25_NO_2_ Na^+^ [M + Na]^+^: 394.1778; Found: 394.1761.

*3-Isopropyl-1-methyl-4-(naphthalen-2-yl)-1-azaspiro[4.5]deca-6,9-diene-2,8-dione* (**4j**). The title compound was prepared according to the general procedure described above by the reaction between *N*-(4-hydroxyphenyl)-*N*-methyl-3-(naphthalen-2-yl)acrylamide (**1j**) with isobutyraldehyde (**2a**), and purified by flash column chromatography as white solid (21.7 mg, 62%). m.p. 84.4–86.8 °C. ^1^H-NMR (400 MHz, CDCl_3_) δ 7.80–7.73 (m, 3H), 7.58–7.54 (m, 1H), 7.50–7.45 (m, 2H), 7.22 (dd, *J* = 8.4, 2.0 Hz, 1H), 6.84 (dd, *J* = 10.2, 3.2 Hz, 1H), 6.63 (dd, *J* = 10.2, 3.0 Hz, 1H), 6.41 (dd, *J* = 10.2, 2.0 Hz, 1H), 5.94 (dd, *J* = 10.2, 2.0 Hz, 1H), 3.61 (d, *J* = 11.8 Hz, 1H), 3.27 (dd, *J* = 11.8, 3.8 Hz, 1H), 2.76 (s, 3H), 2.42–2.34(m, 1H), 1.04 (d, *J* = 6.8 Hz, 3H), 0.83 (d, *J* = 7.0 Hz, 3H). ^13^C-NMR (100 MHz, CDCl_3_) δ 184.38, 175.20, 149.49, 146.96, 133.11, 132.76, 132.40, 131.58, 133.02, 128.46, 127.88, 127.80, 127.46, 126.63, 126.46, 125.83, 65.04, 50.97, 49.34, 28.21, 27.15, 20.25, 18.80. IR (cm^−1^): 3052, 2960, 2931, 2874, 1691, 1672, 1629, 1466, 1418, 1392, 1374, 1261, 1173, 1141, 1064, 992, 854, 822, 735, 607. HRMS: calcd. for C_23_H_23_NO_2_ Na^+^ [M + Na]^+^: 368.1621; Found: 368.1605.

*4-(Furan-2-yl)-3-isopropyl-1-methyl-1-azaspiro[4.5]deca-6,9-diene-2,8-dione* (**4k**). The title compound was prepared according to the general procedure described above by the reaction between 3-(furan-2-yl)-*N*-(4-hydroxyphenyl)-*N*-methylacrylamide (**1k**) with isobutyraldehyde (**2a**), and purified by flash column chromatography as yellow oil (20.5 mg, 72%). ^1^H-NMR (400 MHz, CDCl_3_) δ 7.29–7.28 (m, 1H), 6.70 (dd, *J* = 10.0, 3.1 Hz, 1H), 6.63 (dd, *J* = 10.2, 3.0 Hz, 1H), 6.43 (dd, *J* = 10.0, 2.0 Hz, 1H), 6.24 (dd, *J* = 3.2, 2.0 Hz, 1H), 6.10–6.07 (m, 2H), 3.48 (d, *J* = 11.8 Hz, 1H), 3.17 (dd, *J* = 12.0, 4.0 Hz, 1H), 2.72 (s, 3H), 2.42–2.34 (m, 1H), 0.98 (d, *J* = 6.8 Hz, 3H), 0.86 (d, *J* = 7.2 Hz, 3H). ^13^C-NMR (100 MHz, CDCl_3_) δ 184.51, 174.69, 149.14, 149.04, 146.70, 142.45, 132.12, 131.24, 110.67, 108.59, 64.25, 48.72, 44.06, 27.76, 27.04, 19.70, 18.14. IR (cm^−1^): 3049, 2962, 2933, 2876, 1697, 1674, 1631, 1505, 1420, 1390, 1373, 1261, 1173, 1149, 1118, 1065, 1021, 859, 736, 599. HRMS: calcd. for C_17_H_19_NO_3_ Na^+^ [M + Na]^+^: 308.1257; Found: 308.1248.

*2,6-di-tert-Butyl-4-isopropyl-4-methylcyclohexa-2,5-dienone* (**5**) [52]. The title compound was prepared from mechanistic experiment. ^1^H-NMR (400 MHz, CDCl_3_) δ 6.44 (s, 2H), 1.80–174 (m, 1H), 1.24 (s, 18H), 1.17 (s, 3H), 0.84 (d, *J* = 6.8 Hz, 6H). ^13^ C-NMR (100 MHz, CDCl_3_) δ 186.90, 146.95, 145.80, 42.43, 37.55, 34.89, 29.70, 24.72, 18.09. IR (cm^−1^): 3001, 2961, 2876, 1658, 1643, 1460, 1374, 1267, 1061, 867, 751.

*3-Isopropyl-6-methoxy-1-methyl-4-phenyl-3,4-dihydroquinolin-2(1H)-one* (**6**). The title compound was prepared according to the general procedure described above by the reaction between *N*-(4-methoxyphenyl)-*N*-methylcinnamamide with isobutyraldehyde (**2a**), and purified by flash column chromatography as colorless oil (22.9 mg, 74%). ^1^H-NMR (400 MHz, CDCl3) δ 7.24–7.13 (m, 3H), 7.00–6.97 (m, 3H), 6.85 (dd, J = 8.8, 3.2 Hz, 1H), 6.75 (d, J = 2.8 Hz, 1H), 4.13 (s, 1H), 3.78 (s, 3H), 3.33 (s, 3H), 2.57 (dd, J = 9.2, 2.0 Hz, 1H), 1.70–1.63 (m, 1H), 1.04 (d, J = 6.6 Hz, 3H), 0.98 (d, J = 6.8 Hz, 3H). ^13^C-NMR (100 MHz, CDCl3) δ 170.31, 155.69, 141.99, 133.86, 128.82, 128.23, 127.19, 126.84, 115.88, 115.52, 112.61, 56.64, 55.56, 45.37, 29.73, 28.55, 21.19, 21.05. IR (cm^−1^): 3060, 3025, 2960, 2933, 2872, 2835, 1731, 1666, 1590, 1503, 1469, 1432, 1387, 1341, 1249, 1034, 910, 810, 699, 621. HRMS: calcd. for C_20_H_23_NO_2_ Na+ [M + Na]+: 332.1621; Found: 332.1610.

## 4. Conclusions

We have developed a convenient Fe-catalyzed decarbonylative alkylative spirocyclization of *N*-aryl cinnamamide with aliphatic aldehydes to provide 1-azaspirocyclohexadienones. Readily available aliphatic aldehydes were decarbonylated into primary, secondary and tertiary alkyl radicals readily for the cascade construction of dual C(sp^3^)-C(sp^3^) and C=O bonds. The application of cheap and readily available aliphatic aldehydes as alkyl source, convenient experimental operation for the alkyl radical generation, green ferrous catalyst and reaction solvent, C=C bond difunctionalization strategy and versatile synthetic utilities of the 1-azaspirocyclohexadienone, would render this decarbonylative alkylative spirocyclization attractive for organic synthesis and medicinal chemistry.

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
