# Peer review of "Fe-catalyzed Decarbonylative Alkylative Spirocyclization of N-Arylcinnamamides: Access to Alkylated 1-Azaspirocyclohexadienones"

_molecules, 2020, doi:10.3390/molecules25030432_

Round 1

Reviewer 1 Report

This manuscript is acceptable for publication once the authors have clarified the query stated below.

Q.: The authors depict their resulting products (e.g., 3a in Table 1) as chiral materials. Is this the case or do we have simply trans-addition products as a mixture of R/S and S/R materials, devoid of cis- and or other diastereomeric -forms? Regardless of the answer, details of this need to be included in the text. If they are enantio-pure, how is this measured (chiral GC? optical rotation?). Please clarify.

Reviewer 2 Report

What the authors report herein is a very nice piece of radical chemistry focusing on the development of an interesting novel decarbonylative cyclization strategy to access vAluable spirocyclic compounds. The paper has been  presented in a very illustrative way and the results are absolutely sound. I thus strongly recommend publication of the article after some minor revision!

The comments I have are more related to the presentation of the results than the science described herein. First of all I found Scheme 1 a bit unclear and I recommend that part b (this work) in this Scheme may already contain the target reaction so that this becomes more clear!

Then it would be nice to read abit more about the initial trial to obtain the actual conditions of choice i.e. the authors describe rather special cond. in entry 1 of table 1 already but do not discuss how these cond. where actually obtained?! So a bit more of discussion along this initial screening would be interesting!

I also wonder why the authors apply exactly 122C for the reactions'?!

Please also comment on the diastereoselectivity of products 3 and how the trans config. was assigned!

Finally I wonder if the authors also applied microwave cond.?

But these are really the only question I have to an otherwise very nice paper!
